# Krill Oil Inhibits NLRP3 Inflammasome Activation in the Prevention of the Pathological Injuries of Diabetic Cardiomyopathy

**DOI:** 10.3390/nu14020368

**Published:** 2022-01-15

**Authors:** Xuechun Sun, Xiaodan Sun, Huali Meng, Junduo Wu, Xin Guo, Lei Du, Hao Wu

**Affiliations:** 1Department of Nutrition and Food Hygiene, School of Public Health, Cheeloo College of Medicine, Shandong University, 44 Wenhuaxi Rd., Jinan 250012, China; 201915788@mail.sdu.edu.cn (X.S.); 201915787@mail.sdu.edu.cn (H.M.); xguo@sdu.edu.cn (X.G.); 2Research Center of Translational Medicine, Jinan Central Hospital, Cheeloo College of Medicine, Shandong University, 105 Jiefang Rd., Jinan 250013, China; 3Intensive Care Unit, The Second Hospital, Cheeloo College of Medicine, Shandong University, 247 Beiyuan Rd., Jinan 250033, China; xdsun@email.sdu.edu.cn; 4Department of Cardiology, The Second Hospital of Jilin University, 218 Ziqiang St., Changchun 130041, China; wujd@jlu.edu.cn

**Keywords:** krill oil, NLRP3 inflammasome, diabetes, diabetic cardiomyopathy

## Abstract

Diabetic cardiomyopathy (DCM) is a common complication of diabetes mellitus (DM), resulting in high mortality. Myocardial fibrosis, cardiomyocyte apoptosis and inflammatory cell infiltration are hallmarks of DCM, leading to cardiac dysfunction. To date, few effective approaches have been developed for the intervention of DCM. In the present study, we investigate the effect of krill oil (KO) on the prevention of DCM using a mouse model of DM induced by streptozotocin and a high-fat diet. The diabetic mice developed pathological features, including cardiac fibrosis, apoptosis and inflammatory cell infiltration, the effects of which were remarkably prevented by KO. Mechanistically, KO reversed the DM-induced cardiac expression of profibrotic and proinflammatory genes and attenuated DM-enhanced cardiac oxidative stress. Notably, KO exhibited a potent inhibitory effect on NLR family pyrin domain containing 3 (NLRP3) inflammasome that plays an important role in DCM. Further investigation showed that KO significantly upregulated the expression of Sirtuin 3 (SIRT3) and peroxisome proliferator-activated receptor-γ coactivator 1α (PGC-1α), which are negative regulators of NLRP3. The present study reports for the first time the preventive effect of KO on the pathological injuries of DCM, providing SIRT3, PGC-1α and NLRP3 as molecular targets of KO. This work suggests that KO supplementation may be a viable approach in clinical prevention of DCM.

## 1. Introduction

The prevalence of diabetes mellitus (DM) is increasing dramatically worldwide. The global DM prevalence is estimated to be 10.2% (578 million) by 2030 and 10.9% (700 million) by 2045 [1]. Diabetic cardiomyopathy (DCM) is a common complication of DM, exhibiting pathophysiological abnormalities, including inflammation, oxidative/nitrosative stress, cardiomyocyte apoptosis and accumulation of fibrosis in the heart [2]. Eventually, DCM results in heart failure [3]. The current strategies to prevent DCM are far from satisfactory. Hence, it is important to develop more effective approaches for the prevention of DCM.

Inflammation and oxidative stress are key mechanisms in the pathogenesis of DCM [4]. DM induces the expression of a series of cardiac proinflammatory genes, leading to the infiltration of inflammatory cells in the heart [4]. In addition, DM stimulates the formation of cellular reactive oxygen species and advanced glycation end products that enhance oxidative stress [4]. The DM-induced cardiac inflammation and oxidative stress boost mutually, forming a vicious circle that results in detrimental effects, such as apoptosis of cardiomyocytes and formation of fibrosis [5]. Therefore, targeting inflammation and oxidative stress is a viable strategy for the intervention of DCM.

Omega-3 fatty acids, especially docosahexaenoic acid (DHA) and eicosapentaenoic acid (EPA), are known to have anti-inflammatory and anti-oxidative effects [6]. However, little is known for the effects of DHA and EPA on DCM. To date, there has been only one study reporting that DHA protected against palmitate-induced mitochondrial dysfunction in DCM [7]. Thus, the effects of DHA and EPA on DCM warrants more investigation.

Krill oil (KO), extracted from the *Euphausia superba* (Antarctic krill), is an alternative source of marine omega-3 fatty acids [8]. KO is rich in DHA, EPA and astaxanthin [9]. Notably, the DHA and EPA found in KO are mainly in the form of phospholipids, which are more beneficial compared with the triacylglycerol form [10]. Astaxanthin has anti-inflammatory and anti-oxidative effects, protecting against DM and complications, such as diabetic retinopathy, nephropathy, neuropathy and atherosclerosis [11]. Given the anti-inflammatory and anti-oxidative effects of its major components, KO has potential for prevention of DCM. However, the effect of KO on DCM and the underlying mechanism remain unknown. In the present study, we hypothesize that KO could prevent the pathological injuries of DCM possibly through its anti-inflammatory or antioxidant activities. Therefore, KO is evaluated for its effect on DCM using a mouse model of T2DM.

## 2. Materials and Methods

### 2.1. Animals Housing and Experiments

C57BL/6 male mice were purchased from Charles River Laboratories (Beijing, China) and were housed in the Animal Center of Shandong University at 22 °C, on a 12:12 h light–dark cycle, with free access to a standard AIN-93G diet and tap water. The Institutional Animal Care and Use Committee at Shandong University approved all the experimental procedures (permission number: SYKX20200022).

DM was induced in 8 week old mice by intraperitoneal injection with streptozotocin (STZ, Sigma-Aldrich, Shanghai, China) at 50 mg/kg/body weight (BW) per day, for 5 consecutive days. One week after the last injection, mice with fasting (6 h fast) blood glucose levels above 13.89 mmol/L were considered diabetic [12,13]. The nondiabetic control mice (8-week old) were intraperitoneally injected with sodium citrate (pH 4.5) as the vehicle for STZ. After the confirmation of DM, the diabetic mice were immediately fed either a high-fat diet or a 1.5% KO-containing high-fat diet (the ingredients of experimental diets are shown in Appendix A). The average food intake of mice in the KO-treated group is approximately 2.79 g/d (Appendix A), delivering a dose of KO equivalent to 7 g/d for a 60 kg adult according to the body surface area normalization method [14]. KO was provided by Qingdao Antarctic Weikang Biotechnology Co., Ltd. (Qingdao, China). The composition analysis of KO is summarized in Appendix A. The non-diabetic control mice continuously received the standard AIN-93G diet.

For all the mice, body weight and food intake were recorded every 2 days post DM onset. Blood glucose levels were determined every 4 weeks post DM onset. By the end of the 22nd week post DM, glucose tolerance test (GTT) was performed. After 24 weeks of DM, the mice were euthanized under an aesthesia by intraperitoneal injection of chloral hydrate (0.3 mg/kg), with their hearts harvested for analysis.

### 2.2. Glucose Tolerance Test (GTT)

After a 12 h fast, the mice were intraperitoneally injected with glucose at a dose of 2 g/kg/BW. Blood glucose levels were recorded at 0, 30, 60 and 120 min post the injection.

### 2.3. Assessment of Cardiac Pathology

After harvesting, the heart tissues were fixed into a 10% buffered formalin solution and embedded in paraffin, followed by sectioning into 5 µm thick sections onto glass slides. Hematoxylin and eosin (H&E) staining (ThermoFisher Scientific, Shanghai, China) was performed to evaluate cardiac histology and inflammatory cell infiltration.

Terminal deoxynucleotidyl transferase dUTP nick-end label (TUNEL) staining (KeyGEN BioTECH, Nanjing, China) was performed to detect cardiomyocyte apoptosis. Masson’s trichrome staining (Solarbio, Beijing, China) was used for the evaluation of cardiac fibrosis. The infiltration of inflammatory cells, TUNEL positive cells and Masson’s positive area were quantified using Image J software (National Institutes of Health, Bethesda, MD, USA). The selection of areas to photograph and scoring was conducted by people blind to the identity of the samples.

### 2.4. RNA Isolation and Quantitative Real-Time Polymerase Chain Reaction (qRT-PCR) Analysis

For the determination of mRNA levels, total RNA was extracted from the heart tissue using an RNAeasy™ Animal RNA Isolation Kit (Beyotime, Shanghai, China). RNA concentration and purity were measured using Nanodrop 2000c (ThermoFisher Scientific). The total RNA was reverse transcribed into complementary DNA using a PrimeScript RT reagent Kit (Takara Biomedical Technology, Beijing, China). qRT-PCR was carried out in a 20 μL reaction volume containing 10 μL LightCycler 480 SYBR Green I Master (Roche, Shanghai, China), 1 μL forward primer, 1 μL reverse primer, 2 μL cDNA and 6 μL RNase Free dH_2_O. A PCR was carried out in LightCycler 480II Real-Time PCR system (Roche). The fluorescence intensity of each sample was measured at each temperature change to monitor amplification of the genes. The comparative cycle time (CT) was used to determine fold differences between the samples. Levels of the housekeeping gene acidic ribosomal phosphoprotein P0 (*Rplp0*, also known as *36b4*) were used as an internal control for the normalization of RNA quantity and quality differences among the samples. Fold change in gene expression was normalized to *Rplp0* by the ΔΔCT method using equation 2^−ΔΔCT^. The results were presented as fold changes compared with the control group. The primers for actin alpha 2 (*Actα2*), *Rplp0*, C-C motif chemokine ligand 2 (*Ccl2*), collagen type I alpha 1 (*Col1α1*), gasdermin D (*Gsdmd*), intercellular adhesion molecule 1 (*Icam1*), interferon gamma (*Ifn-γ*), interleukin 1 beta (*Il-1β*), NADPH oxidase 4 (*Nox4*), selectin E (*Sele*), sirtuin 3 (*Sirt3*) and tumor necrosis factor alpha (*Tnf-α*) were synthesized at Sangon Biotech (Shanghai, China). The sequences of the primers are listed in Appendix A.

### 2.5. Western Blot Analysis

To measure protein levels, cardiac tissue was homogenized in a lysis buffer (Beyotime). Western blot analysis was carried out as previously described [15]. Briefly, a bicinchoninic acid assay (Thermofisher Scientific) was used for the determination of the protein concentration. A standard curve was constructed by determining the absorbance of bovine serum albumin with concentrations of 0, 0.025, 0.125, 0.5, 0.75, 1 and 1.5 mg/mL. For the measurement of the absorbance of the protein samples, after adding 200 μL of working solution to each well of a 96-well plate, 25 μL of the protein sample was added to each well, and the absorbance was measured at 562 nm after incubation at 37 °C for 30 min. The protein concentration was then calculated using the standard curve. The primary antibodies used were apoptosis-associated speck-like protein containing a CARD (ASC, 1:1000, Cell Signaling Technology, Shanghai, China), β-tubulin (1:1000, Proteintech, Wuhan, China), Caspase-1 (1:1000, AdipoGen Life Sciences, Beijing, China), collagen type III alpha 1 (COL3A1, 1:1000, Santa Cruz Biotechnology, Shanghai, China), gasdermin D (GSDMD, 1:1000, Cell Signaling Technology), inducible nitric oxide synthase (iNOS, 1:1000, Cell Signaling Technology), IL-1β (1:1000, Cell Signaling Technology), NLR family pyrin domain containing 3 (NLRP3, 1:1000, Cell Signaling Technology) and peroxisome proliferator-activated receptor-γ coactivator 1α (PGC-1α, 1:1000, Affinity Biosciences Ltd., Shanghai, China). Western blot images were quantified utilizing Image StudioTM Lite software (LI-COR, Lincoln, NE, USA).

### 2.6. Immunohistochemical Staining

To determine cardiac protein expression and localization, the tissue sections were deparaffinized and rehydrated. Endogenous peroxidase was inactivated by 3% hydrogen peroxide. Antigens were retrieved using citrate buffer (0.01 M, pH 6.0) at 100 °C for 3 min. After blocking with 5% BSA, the sections were incubated with antibodies against alpha-smooth muscle actin (α-SMA, 1:100, Santa Cruz Biotechnology), COL1A1 (1:100, Santa Cruz Biotechnology), 8-hydroxy-2’-deoxyguanosine (8-OHdG, 1:100, Santa Cruz Biotechnology) or PGC-1α (1:100, Affinity Biosciences Ltd.) at 4 °C overnight. Following incubation with a secondary antibody at 37 °C for 1 h, color was developed with a DAB Horseradish Peroxidase Color Development Kit (BOSTER Biological Technology, Wuhan, China), followed by counterstaining with hematoxylin.

### 2.7. Statistical Analysis

Eight mice per group were studied. The data were expressed as the means ± standard deviation. All the assays were conducted in triplicate. The data were analyzed using one-way analysis of variance (SPSS 19.0). The difference between groups were assessed by the least significant difference (LSD). *p* < 0.05 was considered statistically significant.

## 3. Results

### 3.1. KO Alleviated the DM-Induced Cardiac Pathological Injuries

To investigate the effect of KO on DM-induced cardiac pathological injuries, C57BL/6 mice were induced to DM and were treated with KO for 24 weeks. Blood glucose levels were elevated in the diabetic mice and were not affected by KO (Figure 1A). In consistence, KO did not improve the DM-induced glucose intolerance (Figure 1B,C). Body weight, heart-weight-to-body-weight ratio and heart-weight-to-tibia-length ratio were not affected by KO (Appendix A). The diabetic mice had increased infiltration of inflammatory cells into the paravascular spaces (Figure 1D,E), and developed cardiomyocyte hypertrophy (Figure 1D). These effects were remarkably inhibited by KO (Figure 1D,E). Moreover, KO significantly reduced the DM-enhanced accumulation of cardiac fibrosis (Figure 1F,G) and apoptotic cell death (Figure 1H,I).

### 3.2. KO Prevented the DM-Induced Cardiac Expression of Profibrotic Genes

Since the DM-induced cardiac fibrosis was inhibited by KO, KO was tested for its effect on cardiac profibrotic gene expression. The diabetic mice had increased cardiac expression of *Actα2* and *Col1α1* mRNAs (Figure 2A,B), and COL3A1 protein (Figure 2C). IHC staining revealed increased α-SMA and COL1A1 positive areas in the diabetic heart (Figure 2D,E). These effects were markedly prevented by KO (Figure 2A–E).

### 3.3. KO Attenuated the DM-Generated Cardiac Oxidative Stress

To assess the effect of KO on cardiac oxidative stress, the mRNA levels of *Nox4* (Figure 3A), protein levels of iNOS (Figure 3B), positive stain of DHE (Figure 3C) and 8-OHdG (Figure 3D) were determined. All these oxidative markers were elevated in the DM group and were decreased in the presence of KO (Figure 3A–D).

### 3.4. KO Ameliorated the DM-Promoted Proinflammatory Gene Transcription

Based on the KO-reduced infiltration of inflammatory cells in the hearts of the diabetic mice (Figure 1D), KO was further evaluated for its effect on the transcription of proinflammatory genes. The diabetic mice had increased mRNA levels of *Tnf-α* (Figure 4A), *Ifn-γ* (Figure 4B), *Icam1* (Figure 4C), *Sele* (Figure 4D), *Ccl2* (Figure 4E) and *Il-1β* (Figure 4F), all of which were significantly decreased by KO.

### 3.5. KO Drastically Inhibited NLRP3 Inflammasome Activation in the Diabetic Hearts

To further investigate the mechanism by which KO markedly inhibited cardiac proinflammatory gene transcription in the diabetic mice (Figure 4A–F), the protein levels of NLRP3 inflammasome components—NLRP3 (Figure 5A), cleaved caspase-1 (Figure 5B), ASC (Figure 5C) and cleaved IL-1β (Figure 5D)—were measured. All these proteins were increased in the DM group and were dramatically reduced by KO (Figure 5A–D). As cleavage of GSDMD is induced by cleaved caspase-1, leading to pyroptosis in DCM [16], we further determined the protein levels of cleaved GSDMD (Figure 5E) and mRNA levels of *Gsdmd* (Figure 5F), both of which were increased in the diabetic hearts and were decreased by KO (Figure 5E,F).

### 3.6. KO Activated Cardiac PGC1-α/SIRT3 as the Upstream Regulators of NLRP3

In order to further explore potential molecular targets of KO, we determined the expression of PGC1-α/SIRT3 that are known as the upstream regulators of NLRP3 in kidneys of the mice. KO significantly increased both mRNA and protein levels of cardiac SIRT3 (Figure 6A,B) and protein levels of cardiac PGC1-α (Figure 6C) in the diabetic mice. IHC staining further confirmed that PGC1-α protein was reduced in the diabetic cardiomyopathy, whereas KO could increase cardiac PGC1-α protein expression (Figure 6D).

## 4. Discussion

In the present study, we report the preventive effect of KO on the pathological injuries of DCM using a mouse model of T2DM. The diabetic mice developed cardiac pathological injuries, inflammation and oxidative stress, the effects of which were prevented by KO. Notably, KO significantly inhibited NLRP3. Further investigation revealed SIRT3/PGC-1α as potential molecular targets of KO (Figure 7).

Accumulation of fibrosis and apoptosis of cardiomyocytes are key pathological features of DCM [17]. In the DM milieu, extracellular matrix proteins are overproduced and form fibrosis in the heart, contributing to cardiac dysfunction [4,17,18]. Apoptosis of cardiomyocytes causes a reduction in the cardiomyocyte population, enhancing the burden of viable cardiomyocytes to maintain normal cardiac function. This subsequently enlarges the cells, leading to pathological hypertrophy and the following cardiac dysfunction [19]. Therefore, the inhibition of profibrotic gene expression and cardiomyocyte apoptosis is key to the successful prevention of DCM. In the present work, we observed the potent inhibitory effects of KO on DM-induced cardiac fibrosis and apoptotic cell death (Figure 1F–I and Figure 2), shedding light on the future clinical prevention of DCM.

Inflammation and oxidative stress cause the generation of fibrosis and apoptosis of cardiomyocytes, thereby playing essential roles in the pathogenesis of DCM [20]. DM-induced cardiac inflammation is driven by NLRP3 inflammasome [21,22], which can be activated by reactive oxygen species (ROS) [23]. Upon various stimuli, including ROS, pro-caspase-1 is cleaved, resulting in the cleavage of proinflammatory cytokines IL-1β and IL-18, and the protein gasdermin D (GSDMD). The N-terminal domain of GSDMD penetrates pores in the plasma membrane, thereby triggering a lytic, pro-inflammatory form of cell death, namely pyroptosis [16]. Pyroptosis leads to the release of cytokines that, in turn, exacerbate NLRP3-mediated inflammation and facilitate inflammatory cells infiltration [24]. In the present study, we found remarkable elevated cardiac oxidative stress and proinflammatory gene transcription in the diabetic mice (Figure 3 and Figure 4). In line with previous reports [23,25,26], NLRP3 was significantly activated in the diabetic hearts (Figure 5A–D). Notably, KO had a striking inhibitory effect on the DM-activated NLRP3 (Figure 5A–D). These data suggest that NLRP3 is an important molecular target of KO.

Recent findings have uncovered important protective roles of SIRT3 and PGC-1α in DCM [27,28,29]. SIRT3 belongs to the sirtuin family of proteins, exerting an NAD+-dependent deacetylase activity [30]. Compared with wild-type (WT) diabetic mice, Sirt3 gene knockout diabetic mice developed more severe DM-induced cardiac injuries [27]. The activation of SIRT3 by polydatin, icariin and salidroside attenuated DCM in WT diabetic mice [27,29,31]. SIRT3 deacetylase activity is required for the protein stability of PGC-1α [32]—a member of a family of transcription coactivators that plays a central role in the regulation of cellular energy metabolism [33]. PGC-1α, in turn, activates SIRT3 [34,35], and is protective in DCM pathogenesis [36,37,38,39]. Collectively, these studies have demonstrated a positive feedback interaction between SIRT3 and PGC-1α, providing SIRT3/PGC-1α activation as a viable strategy for the intervention of DCM. Moreover, SIRT3 has been reported to be a crucial negative regulator of NLRP3 [40,41,42,43]. Mechanistically, SIRT3 deacetylates SOD2, leading to SOD2 activation, which impairs NLRP3 inflammasome assembly and activation [41]. The present study found that KO upregulated the protein levels SIRT3 and PGC-1α (Figure 6), the effects of which were in accordance with KO’s inhibitory effect on NLRP3. Thus, KO might inhibit NLRP3 via the upregulation of SIRT3/PGC-1α in DCM. The identification of PGC-1α and SIRT3 in addition to NLRP3 provided more viable molecular targets of KO in the prevention of DCM. Although KO is known to have inflammatory efficacies, its effect on NLRP3, SIRT3 and PGC-1α is not previously reported. To date, this has been the first study to report KO’s effect on SIRT3/PGC-1α/NLRP3. Given that SIRT3, PGC-1α and NLRP3 play important roles in various diseases, the findings in our study might indicate SIRT3, PGC-1α and NLRP3 as potential mechanisms through which KO benefits diseases.

In human studies, KO has proven beneficial for a few diseases, including osteoarthritis, arthritis, knee joint pain and hyperlipidemia [44], all of which are closely associated with inflammation and oxidative stress. To date, KO has not been investigated for its effects on clinical DM and complications. The potent inhibitory effects of KO on DM-induced cardiac inflammation holds promise for future clinical intervention of DM and complications, including DCM. Additionally, the present work might indicate that functional lipids containing bioactive components, such as KO, hazelnut oil [45], among others, warrant more attention in the future intervention of diseases.

## 5. Conclusions

The present study reports for the first time the preventive effect of KO on the pathological injuries of DCM. In addition, SIRT3, PGC-1α and NLRP3 were identified as molecular targets of KO. Our work presents KO supplementation as a viable approach in the clinical intervention of DCM.

## Figures and Tables

**Figure 1 nutrients-14-00368-f001:**
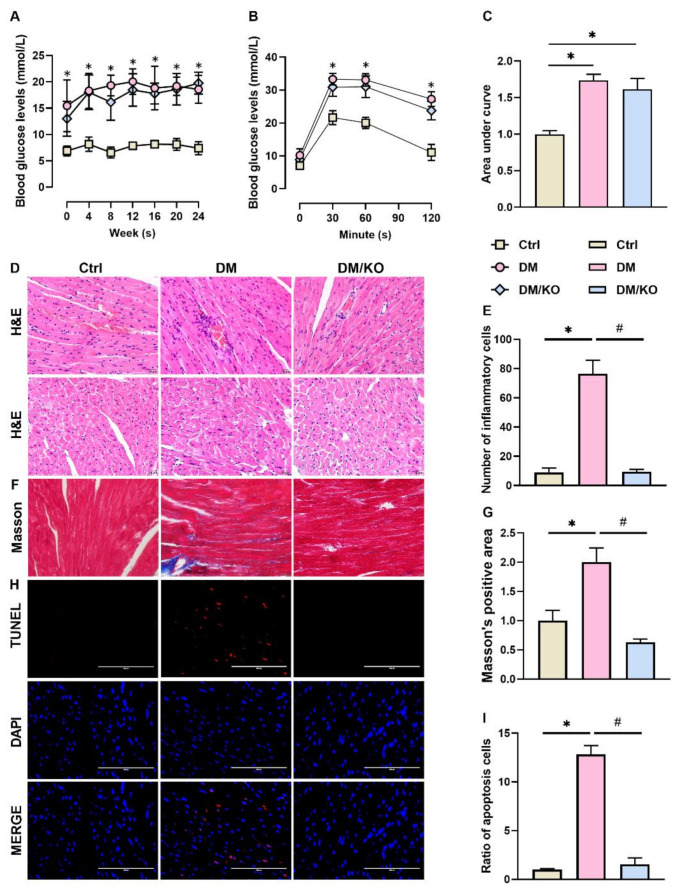
KO alleviated the DM-induced cardiac pathological injuries. (**A**) Blood glucose levels of C57BL/6 male mice were recorded every 4 weeks post diabetes mellitus (DM) onset. (**B**) Glucose tolerance test was performed 22 weeks post DM onset, with (**C**) the areas under the curve quantified. (**D**) Hematoxylin and eosin (**E**,**H**) staining with (**E**) the number of inflammatory cells infiltrated per vessel quantified (Bar = 20 μm). (**F**) Masson’s trichrome staining with (**G**) the positive area quantified (Bar = 20 μm). (**H**) TUNEL staining with (**I**) the ratio of apoptosis cells quantified (Bar = 100 μm). The data were normalized to Ctrl and summarized as means ± SD. *, *p* < 0.05 vs. Ctrl; #, *p* < 0.05 vs. DM. Abbreviations, Ctrl, control; DM, diabetes mellitus; H&E, hematoxylin and eosin; KO, krill oil; TUNEL, terminal deoxynucleotidyl transferase dUTP nick-end label.

**Figure 2 nutrients-14-00368-f002:**
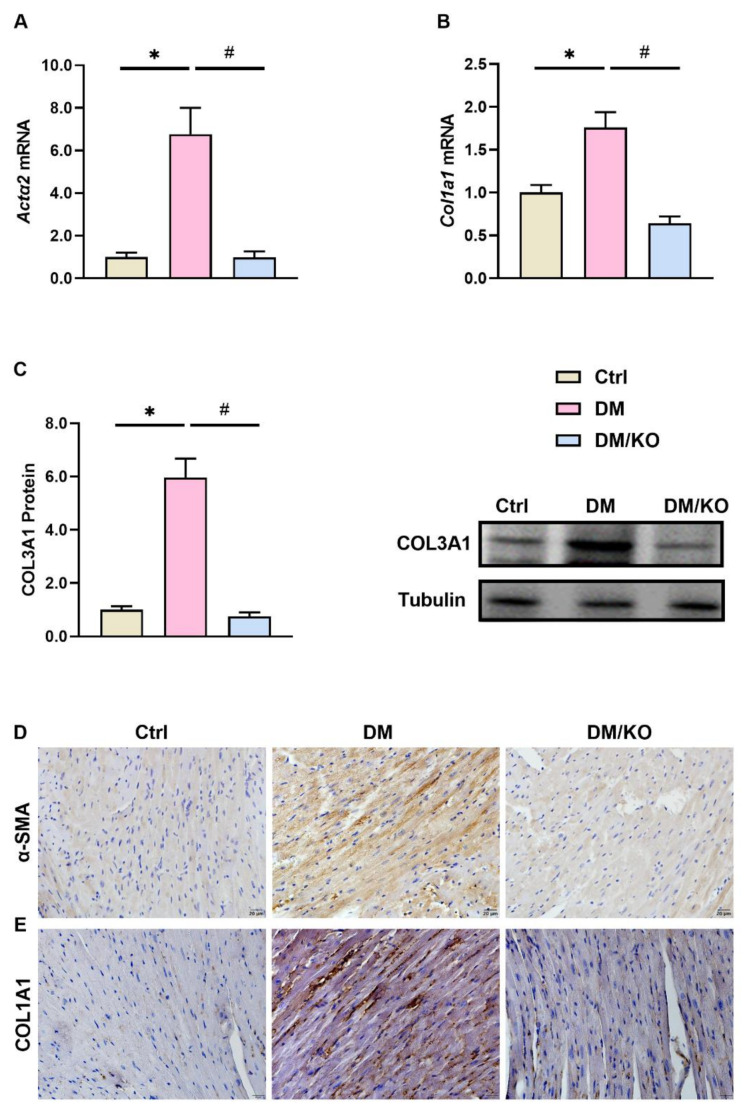
KO prevented the DM-induced cardiac expression of profibrotic genes. Cardiac mRNA expression of (**A**) *Actα2* and (**B**) *Col1α1* were determined by qRT-PCR. Cardiac protein expression of (**C**) COL3A1 was determined by Western blot and (**D**) α-SMA and (**E**) COL1A1 were detected by Immunohistochemical staining (Bar = 20 μm). Tubulin and *Rplp0* (also known as *36b4*) were used as endogenous controls for Western blot and qRT-PCR, respectively. The data were normalized to Ctrl and summarized as means ± SD. *, *p* < 0.05 vs. Ctrl; #, *p* < 0.05 vs. DM. Abbreviations: α-SMA, alpha-smooth muscle actin; *Actα2*, actin alpha 2; *Col1α1*, collagen 1 alpha 1; COL3A1, collagen 3 alpha 1; *Rplp0*, acidic ribosomal phosphoprotein P0. Other abbreviations are the same as Figure 1.

**Figure 3 nutrients-14-00368-f003:**
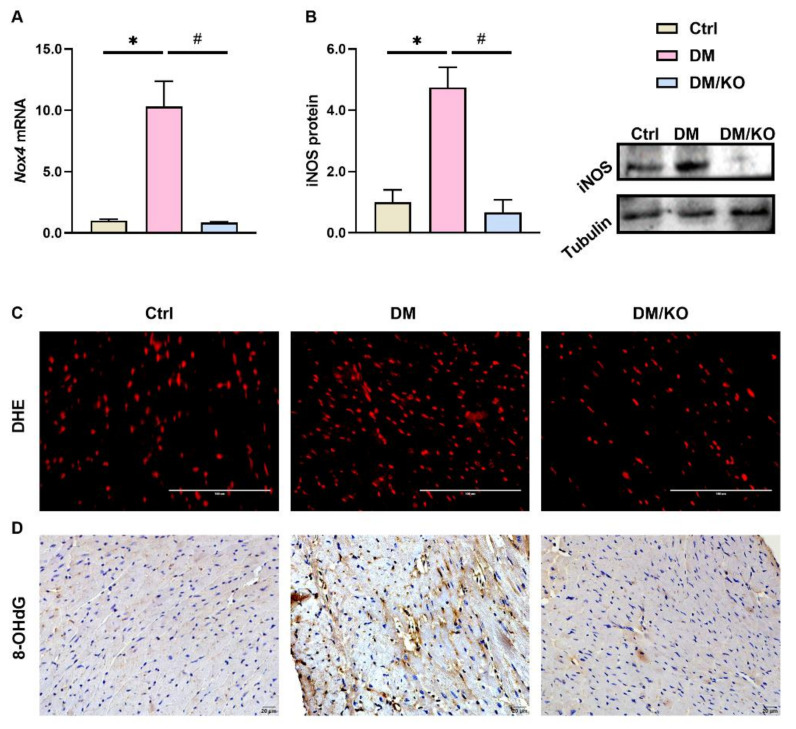
KO attenuated the DM-generated cardiac oxidative stress. (**A**) *Nox4* mRNA levels. (**B**) iNOS protein levels. Tubulin and *Rplp0* were used as endogenous controls for Western blot and qRT-PCR, respectively. (**C**) Representative microphotographs of DHE staining in heart sections in each group (Bar = 100 μm). (**D**) Immunohistochemical staining showed expression of 8-OHdG (Bar = 20 μm). The data were normalized to Ctrl and summarized as means ± SD. *, *p* < 0.05 vs. Ctrl; #, *p* < 0.05 vs. DM. Abbreviations: DHE, dihydroethidium; *Nox4*, NADPH oxidase 4; iNOS, inducible nitric oxide synthase; 8-OHdG, 8-hydroxy-2 deoxyguanosine. Other abbreviations are the same as Figure 1 and Figure 2.

**Figure 4 nutrients-14-00368-f004:**
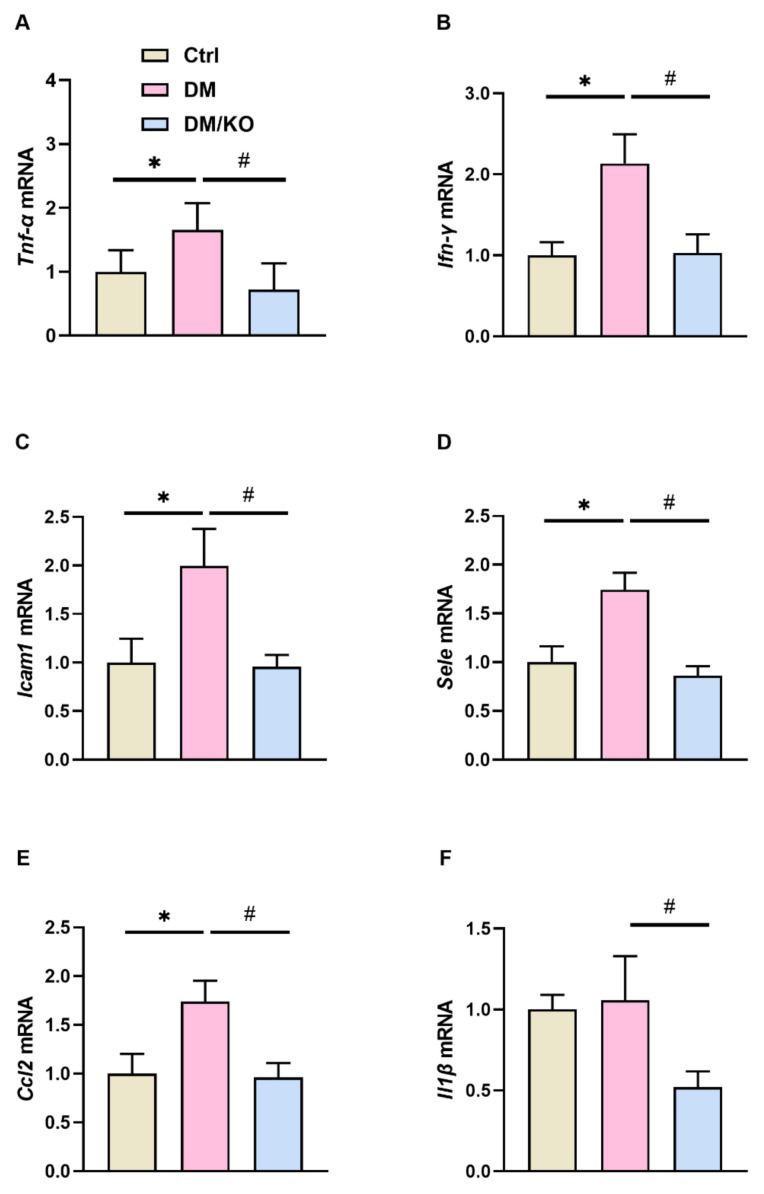
KO ameliorated the DM-promoted proinflammatory gene transcription. Cardiac mRNA expression of (**A**) *Tnf-α*, (**B**) *Ifn-γ*, (**C**) *Icam1*, (**D**) *Sele*, (**E**) *Ccl2*, and (**F**) *Il-1β* were determined by qRT-PCR. *Rplp0* was used as an endogenous control. The data were normalized to Ctrl and summarized as means ± SD. *, *p* < 0.05 vs. Ctrl; #, *p* < 0.05 vs. DM. Abbreviations: *Ccl2*, C–C motif chemokine ligand 2; Icam1, intercellular adhesion molecule 1; *Ifn-γ*, interferon gamma; *Il-1β*, interleukin 1beta; *Sele*, selectin; *Tnf-α*, tumor necrosis factor alpha. Other abbreviations are the same as Figure 1 and Figure 2.

**Figure 5 nutrients-14-00368-f005:**
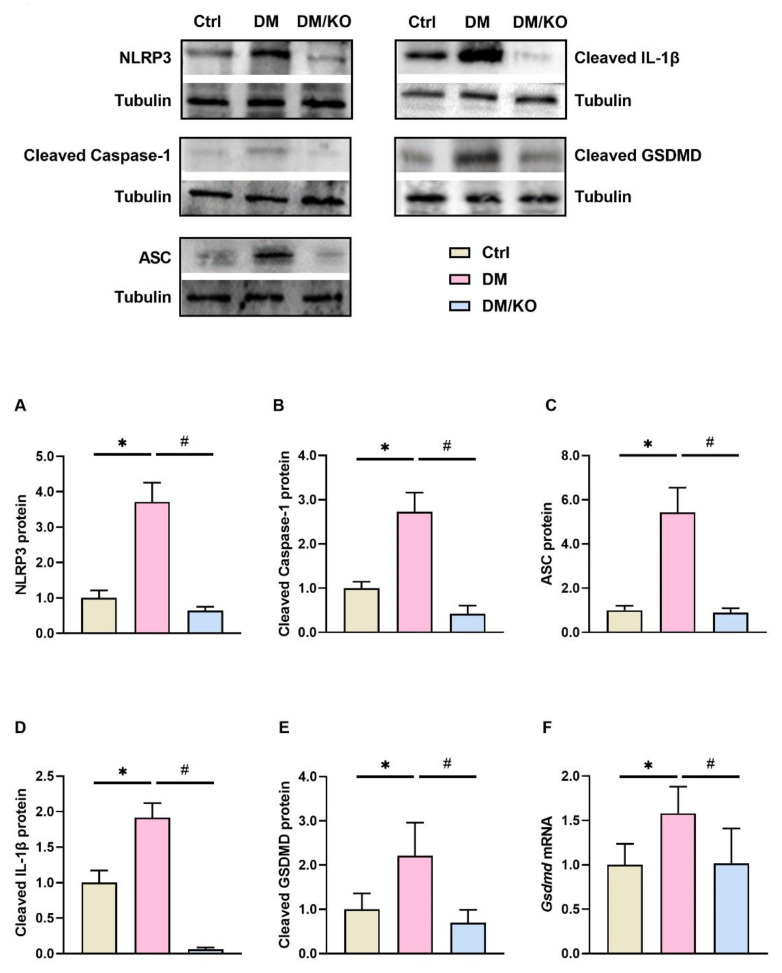
KO drastically inhibited NLRP3 inflammasome activation in the diabetic hearts. Cardiac protein expression of (**A**) NLRP3, (**B**) Cleaved caspase-1, (**C**) ASC, (**D**) Cleaved IL-1β, (**E**) Cleaved GSDMD were determined by Western Blot. (**F**) *Gsdmd* mRNA levels. Tubulin and *Rplp0* were used as endogenous controls for Western blot and qRT-PCR, respectively. The data were normalized to Ctrl and summarized as means ± SD. *, *p* < 0.05 vs. Ctrl; #, *p* < 0.05 vs. DM. Abbreviations: ASC, adaptor apoptosis-associated speck-like protein containing a caspase activation and recruitment domain; GSDMD, gasdermin D; NLRP3, Nod-like receptor family, pyrin domain containing 3. Other abbreviations are the same as Figure 1 and Figure 2.

**Figure 6 nutrients-14-00368-f006:**
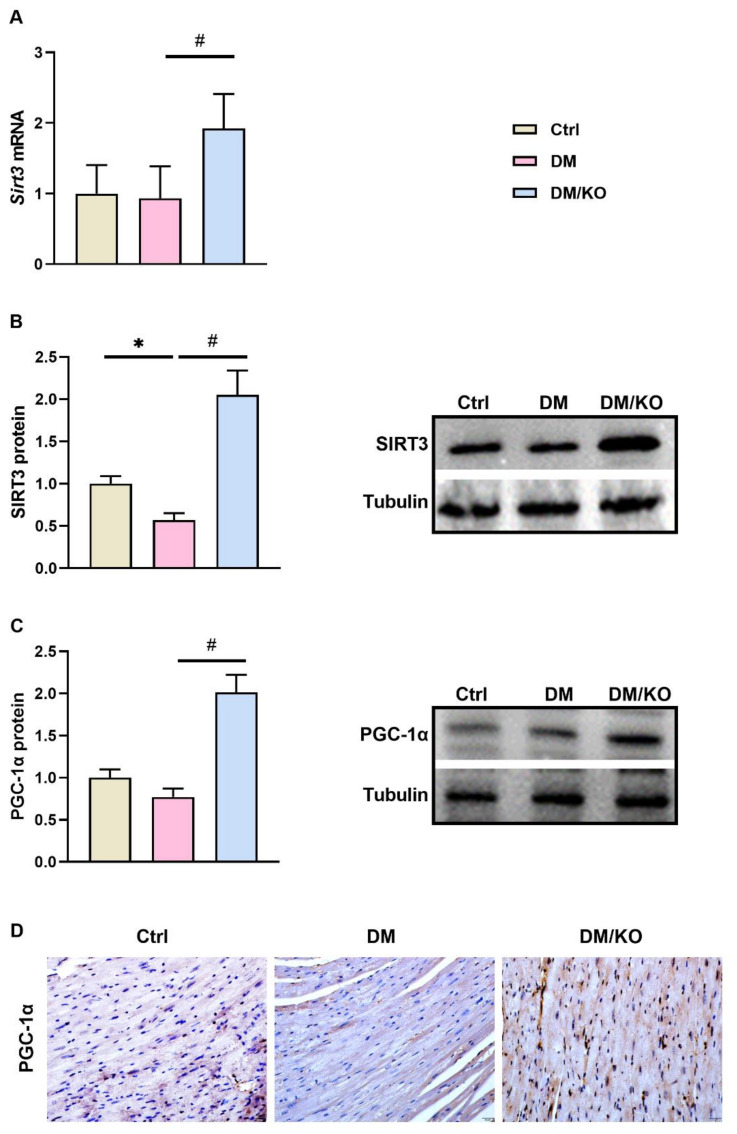
KO activated cardiac PGC1-α/SIRT3 as the upstream regulators of NLRP3. (**A**) *Sirt3* mRNA levels, (**B**) SIRT3 protein levels, (**C**) PGC-1α protein levels, (**D**) immunohistochemical staining of cardiac PGC-1α (Bar = 20 μm). Tubulin and *Rplp0* were used as endogenous controls for Western blot and qRT-PCR, respectively. The data were normalized to Ctrl and summarized as means ± SD. *, *p* < 0.05 vs. Ctrl; #, *p* < 0.05 vs. DM. Abbreviations: PGC-1α, PPARG coactivator 1 alpha; SIRT3, sirtuin 3. Other abbreviations are the same as Figure 1 and Figure 2.

**Figure 7 nutrients-14-00368-f007:**
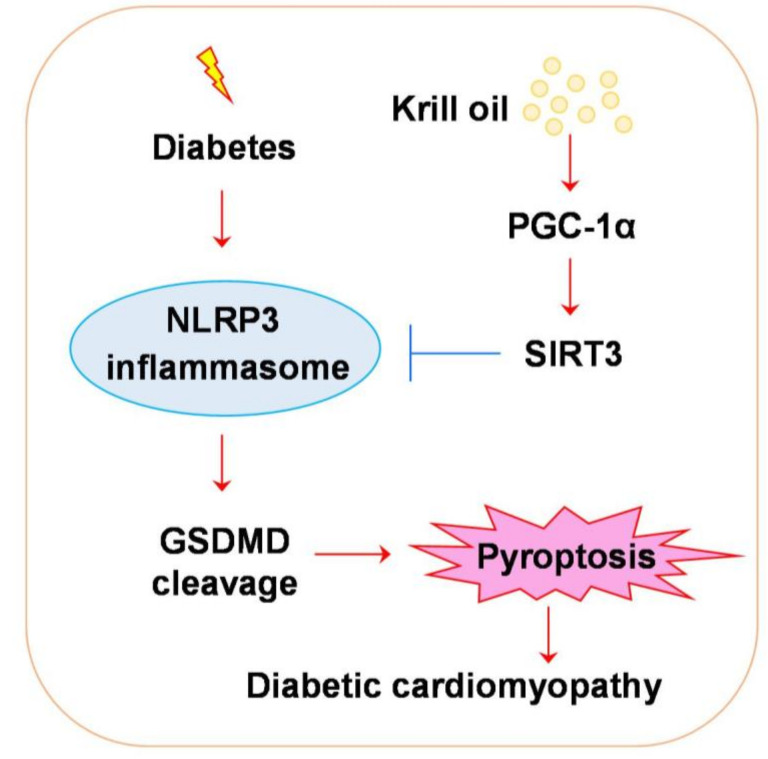
The possible molecular mechanism of KO in inhibiting pyroptosis in DCM. Under diabetic condition, NLRP3 inflammasome is activated, leading to the cleavage of GSDMD and the following pyroptosis. KO could upregulate the expression of PGC-1α and SIRT3, which are upstream negative regulators of NLRP3 inflammasome. ↓, activation; ┴, inhibition. Abbreviations: DCM, diabetic cardiomyopathy. Other abbreviations are the same as in Figure 1, Figure 5 and Figure 6.

## Data Availability

The datasets used and/or analyzed during the current study are available from the corresponding author on reasonable request.

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
