# Peer review of "Krill Oil Inhibits NLRP3 Inflammasome Activation in the Prevention of the Pathological Injuries of Diabetic Cardiomyopathy"

_nutrients, 2022, doi:10.3390/nu14020368_

Round 1

Reviewer 1 Report

In this study, the Authors investigated the effects krill oil  on prevention of diabetic cardiomyopathy using a diabetic (streptozotocin-induced ) murine model fed high-fat diet. Their findings might be insightful  for the clinical arena in the context of diabetic cardiomyopathy.

The work has merit, it is narrowly conducted, and well structured.

The design is sound although a power analysis (why 8 animals per group) would have been of assistance.

On a singular note, the introduction would benefit from citing similar papers on such model/purposes:

  • 10.1016/j.dld.2019.09.002
  • 10.5650/jos.ess18086

Reviewer 2 Report

The presented studies may be interesting for readers, as it raises an important issue concerning the effect of KO on pathological injuries of DCM in a mouse model of T2DM. Despite the fact that the work seems to be properly conducted, due to some comments about it, as presented in feedback for Authors, the manuscript should be considered for publication in Nutrients after minor revision.

Please, respond to the remarks as below:

  1. In the Introduction section, the purpose of the research has not been clearly described.
    Lines 65-66: please specify the purpose and the underlining hypothesis of the study.

  1. Materials and Methods section

Interspecies allometric scaling for dose conversion from animal to human studies and converesely is one of the most controversial areas in clinical pharmacology. Please describe in details your KO dose calculations (including the body surface area correction factor, scaling factors from human), and include them in the text (Lines 81-84).

Lines 88-92: this part of the study description is not clear. Explain exactly for which group of mice (only for DM?) the described procedures were used?

Incomplete description of the Table S3 in the Suplementary Material file. Missing GenBank accession number and amplicon size of primers.

Describe in more detail the qRT-PCR analysis. Lack of information about validation of primers using a positive control, final concetrations of the target/reference genes used in the assay, conditions of amplification, etc.

Line 128: Please describe the protein concentration determination method used for the Western blot assay.

Lines 153-156: Lack of the description of the calculation of the mRNA expression levels, what post-hoc test was used, what statistical software was used in the study.

  1. The p-value is missing in the description of the results (Results section), what makes difficult to determine the significance of the results.
  2. The results are well discussed in the Discussion section.
